# Lymphedema Rehabilitation Using Self-Adaptive Inelastic Compression in Breast Cancer: A Proof-of-Principle Study

**Alessandro de Sire** [1,2,*], **Nicola Fusco** [3,4], **Elham Sajjadi** [3,4], **Lorenzo Lippi** [1], **Carlo Cisari** [1,5] and **Marco Invernizzi** [1,6,*]

1 Physical and Rehabilitative Medicine, Department of Health Sciences, University of Eastern Piedmont, 28100 Novara, Italy; lorenzolippi.mt@gmail.com (L.L.); carlo.cisari@med.uniupo.it (C.C.)
2 Rehabilitation Unit, Mons. L. Novarese Hospital, 13040 Vercelli, Italy
3 Department of Oncology and Hemato-Oncology, University of Milan, 20122 Milan, Italy; nicola.fusco@unimi.it (N.F.); elham.sajjadi@unimi.it (E.S.)
4 Division of Pathology, IEO, European Institute of Oncology IRCCS, 20144 Milan, Italy
5 Physical Medicine and Rehabilitation Unit, University Hospital "Maggiore della Carità", 28100 Novara, Italy
6 Infrastruttura Ricerca Formazione Innovazione (IRFI), Azienda Ospedaliera SS. Antonio e Biagio e Cesare Arrigo, 15121 Alessandria, Italy
* Correspondence: alessandro.desire@uniupo.it (A.d.S.); marco.invernizzi@med.uniupo.it (M.I.); Tel.: +39-03213734800

**Abstract:** Inelastic compression (IC) has been proposed as a therapeutic option in the management of breast cancer-related lymphedema (BCRL). To date, no studies have investigated the reliability of IC in the rehabilitative management of BCRL. Thus, we aimed at evaluating the safety and tolerability of an instantly adjustable IC device for BCRL patients. We included BCRL patients referred to an Oncological Rehabilitation Unit. They were subjected to complex decongestion therapy using a self-adaptive IC device. The primary outcome was safety and tolerability of IC in the rehabilitation treatment of BCRL patients. Secondary outcomes were: BCRL volume modifications; disability; quality of life; IC application time. Outcomes were assessed at the baseline (T0), at the end of the 2-week rehabilitation treatment (T1), and at 1 month (T2). Nine BCRL women, mean aged $56.7 \pm 4.4$ years, were enrolled. None showed side effects, except for a low discomfort and moderate swelling reported by 1 patient (11.1%). BCRL volume significantly reduced at T1 ($p = 0.001$) with a positive trend at T2. IC timing was significantly reduced at T1 ($p = 0.035$) and T2 ($p = 0.003$). Taken together, these findings suggested that IC could be considered as a safe, well-tolerated, and effective tool in the rehabilitative management of BCRL patients.

**Keywords:** breast cancer-related lymphedema; lymphedema; rehabilitation; breast cancer; inelastic compression; quality of life

## 1. Introduction

Substantial improvements in the diagnosis and treatment of breast cancer have significantly increased the life expectancy of these women worldwide [1]. However, breast cancer survivors might suffer from a wide range of sequelae, including fatigue [2], axillary web syndrome [3,4], treatment-induced bone loss [5,6], psychological impact [7], and breast cancer-related lymphedema (BCRL) [8].

BCRL is characterized by tissue swelling and subsequent irreversible fibrosis due to abnormal lymph drainage. This condition occurs within two years after treatment in approximately 20% of breast cancer patients [9]. BCRL pathogenesis is mainly related to axillary lymphadenectomy followed by radiotherapy; however, radical breast surgery, high body mass index, socioeconomic factors, and tumor-specific pathologic features are also implicated in its pathogenesis [10,11]. A variety of clinical manifestations including discomfort, pain, cutaneous alteration, upper limb functional impairment, and psychological sequelae, is responsible for a decreased health-related quality of life (HRQoL) in these

patients [12]. Thus, an early diagnosis of BCRL through the circumferential method [13], water displacement [14], or three-dimensional laser scanner [15,16], is currently considered the most reliable strategy for an optimal rehabilitation plan.

In this context, complex decongestive therapy (CDT) is one of the most effective BCRL therapeutic and rehabilitative interventions. This treatment consists of skin hygiene, rehabilitative exercise, manual lymphatic drainage (MLD), and multilayered lymphatic bandaging (MLB) treatment [17–20]. It has been recently shown that MLD combined with standard therapy might enhance the effectiveness of treating volume reduction in BCRL, without improving subjective symptoms or upper limb function [21]. A recent Cochrane Review [22] showed that MLD combined with MLB might be an effective therapeutic strategy in patients with a diagnosis of mild to moderate BCRL. In this respect, the National Lymphedema Network Medical Advisory Committee recommends the use of the least compressive contention during physical exercise in BCRL women [23]. Such devices might help in maintaining the results obtained during CDT rehabilitative sessions [24–26]. Lately, inelastic compression (IC) has been proposed as a therapeutic alternative in combination with MLD in the clinical rehabilitative management of BCRL. This novel approach is based on self-adaption and self-application of IC wraps/garments that allow for increased patient compliance to the treatment.

To date, no studies have currently been performed to evaluate the clinical reliability of IC in the real-life rehabilitation management of BCRL. Here, we sought to evaluate the safety and tolerability of an instantly adjustable IC device in the rehabilitative management of BCRL. Moreover, upper limb volume and disability reduction, along with HRQoL improvement have been assessed.

## 2. Materials and Methods

### 2.1. Participants

We included BCRL patients (Stage II-III), having undergone breast surgery and unilateral lymphadenectomy, that were referred to the Oncological Rehabilitation Unit of the University Hospital "Maggiore della Carità" in Novara, Italy for 6 months, i.e., from August 2019 to January 2020.

The exclusion criteria were the following: (a) age <18 years; (b) first clinical evaluation after >2 weeks from surgery to exclude other complications; (c) skin lesions; (d) trauma or other lesions able to modify the structure or volume of the limb; (e) active phase of the BC or other active malignant tumors; (f) the presence of systemic metastases; (g) radio or chemotherapy cycles during the study; (h) vascular disease (i.e., deep venous thrombosis or superficial thrombophlebitis); (i) previous surgery for BC; (j) unavailability of therapeutic data.

The study was compliant with the ethical guidelines of the responsible governmental agency and was approved by the local Institutional Review Board. All researchers involved were instructed to protect the participants' privacy, and the procedures were performed according to the Declaration of Helsinki.

### 2.2. Inelastic Compression

All patients were subjected to complex physical decongestion therapy using a self-adaptive IC device (Circaid®, Medi GmbH & Co. KG, Bayreuth, Germany). The wraps, which are made of synthetic fibers as nylon (polyamide), elastane, polyurethane, and polyethylene, were cut and sewn according to the shape of the arm. The thickness of the material varied according to the pathology target, e.g., venous ulcers or lymphedema, where thicker wraps were recommended for the latter condition. The IC device employed in this study has a built-in pressure system patented technology that allowed patients to define and control an adequate tension of the bandage.

In this study, patients were instructed by a physical therapist experienced in CDT to wear the self-adaptive IC device (Figure 1).

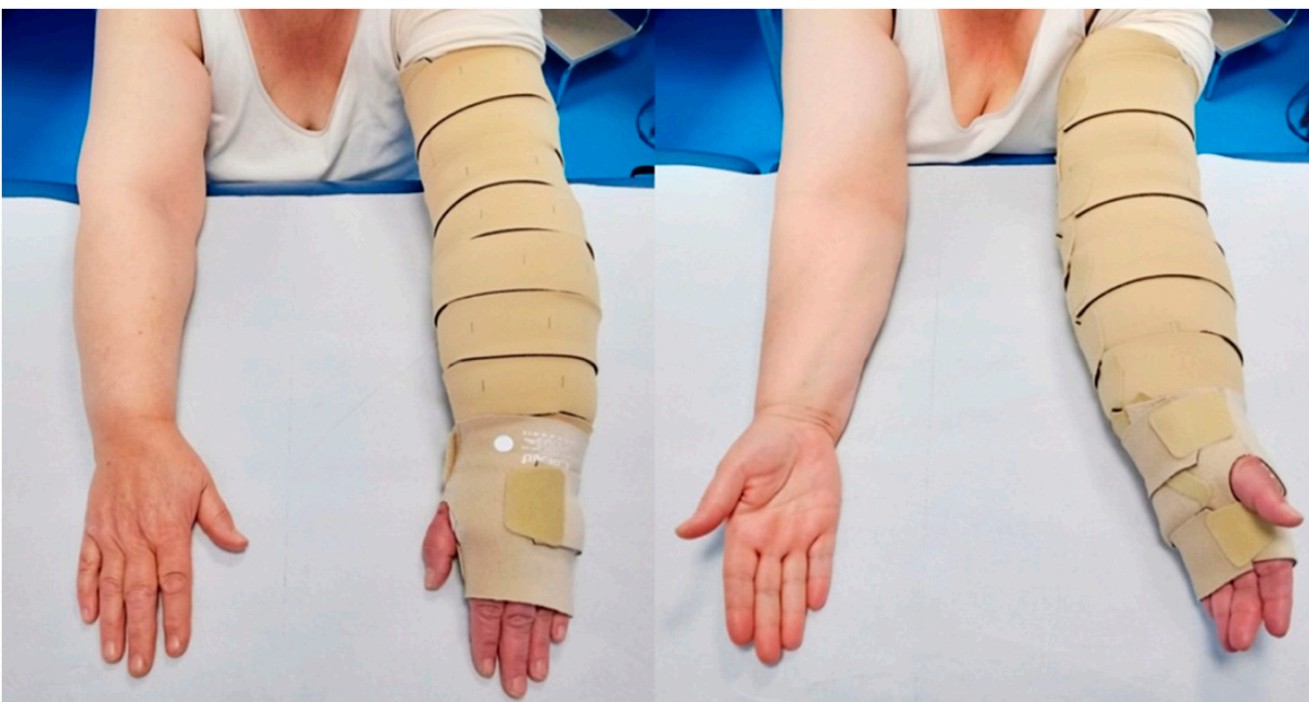

**Figure 1.** Patient with breast cancer related lymphedema wearing the self-adaptive inelastic compression device that was used in this study.

### 2.3. Intervention

All patients underwent a clinical assessment by a physiatrist experienced in cancer rehabilitation. The intervention consisted of a 2-week rehabilitative treatment (5 sessions/week) under the supervision of a physical therapist experienced in CDT.

Each session consisted of: (a) accurate skin hygiene; (b) MLD for 30 min; (c) IC application; (d) 20-min upper limb therapeutic exercise; (e) IC repositioning. Patients wore the IC wraps for 24 h after the first 4 sessions and 48 h after the last session. Lastly, at the end of the 2-week treatment, all patients were asked to perform a 2-week home-based rehabilitative treatment, consisting of the same therapeutic exercises learned during the sessions (except for MLD), monitored through phone calls by verbally evaluating ability to perform exercises by the physical therapist. One month after the first evaluation, all subjects underwent a follow-up physiatric evaluation.

### 2.4. Outcome Measures

The primary outcome consisted of the IC safety and tolerability evaluation. Secondary outcomes included BCRL volume modifications (even considering hand, forearm, and arm subsections), evaluated through circumferential measurement (CM), consisting of measuring through tape with 1 mm of sensitivity upper limb circumferences in correspondence of markers made on the skin from wrist to deltoid muscle level, with 5 cm intervals [27]. The measurements were performed according to standard markers drawn on the skin of the patients, from wrist to deltoid muscle level. Physical, functional, and psychosocial concerns were pertinent to patients with lymphedema, assessed by the Lymphedema Life Impact Scale (LLIS) [28]. Quality of life was assessed by the European Organization for Research and Treatment of Cancer quality of life questionnaire (EORTC QLQ-C30) and its subsections (i.e., function, symptom, and global health) [29]. IC wraps application timing was measured in seconds. All these outcomes were assessed at the first evaluation at the Oncological Rehabilitation Unit (T0, baseline), at the end of the 2-week rehabilitation treatment (T1), and the 1-month follow-up evaluation (T2). Lastly, the enrolled patients

expressed their satisfaction regarding the treatment at T1 and at T2, through the Global perceived effect (GPE) Scale, ranging from 1 (best satisfaction) to 7 (unsatisfaction) [30].

### 2.5. Statistical Analysis

The statistical analysis was performed using GraphPad Prism® software (Version 7.00 GraphPad Software Inc., La Jolla, CA, USA). Continuous variables were expressed as means and standard deviations, whereas the categorical variables were expressed as absolute numbers. The Shapiro-Wilk test was used to assess the data distribution; thus, because data did not follow a normal distribution, differences in repeated measures were assessed by the Friedman Test (one-way ANOVA) with post-hoc Dunnett's Test as a multiple comparison procedure. $p$-values < 0.05 were considered as statistically significant.

## 3. Results

Nine women affected by BCRL, mean age $56.7 \pm 4.4$ years, were enrolled. Four patients were affected by left upper limb BCRL, whereas 5 patients were affected by right upper limb BCRL. All participants underwent breast surgery (mastectomy, n = 3; quadrantectomy, n = 6) with axillary lymph node dissection, as shown in Table 1.

**Table 1.** Baseline characteristics of study participants.

| | |
|---|---|
| Age (years) | $56.7 \pm 4.4$ |
| BMI (kg/m$^2$) | $32.03 \pm 4.07$ |
| BCRL arm involvement | |
| Right upper limb | 5 |
| Left upper limb | 4 |
| Breast cancer surgery | |
| Mastectomy | 3 |
| Quadrantectomy | 6 |
| Axillary lymph node dissection | 9 |
| Pharmacological therapy | |
| Hormonal therapy | 1 |
| Beta-blockers | 3 |
| Anti-depressive drugs | 1 |

Continuous variables are expressed as means ± standard deviations. Categorical variables are expressed as counts. Abbreviations: BMI = body mass index; BCRL = breast cancer related lymphedema.

There were no dropouts in the entire cohort at all the time-points; all patients showed no side effects, except for a low discomfort at the hand and a moderate swelling reported by 1 patient (11.1%). However, it has to be noted that this adverse event was self-limiting and related to incorrect device positioning and did not result in a drop-out. These promising findings in terms of tolerability are confirmed by a GPE score of 1.9 at T1 and 2.7 at T2 for both patients and operators.

There was a statistically significant reduction in terms of upper limb total volume at T1 ($3881.4 \pm 886.9$ cm$^3$ vs $3521.9 \pm 759.2$ cm$^3$; $p = 0.001$) and a positive trend also at T2 ($3521.9 \pm 759.2$ cm$^3$ vs $3650.9 \pm 835.5$ cm$^3$; $p > 0.05$), albeit only hand volume was significantly reduced at T2 ($348.1 \pm 99.0$ cm$^3$ vs $298.6 \pm 80.8$ cm$^3$; $p = 0.007$).

However, regarding disability and HRQoL assessment, there were no significant differences in terms of LLIS score at T1 ($38.9 \pm 10.1$ vs $34.7 \pm 8.1$; $p > 0.05$) and T2 ($34.7 \pm 8.1$ vs $35.1 \pm 12.6$; $p > 0.05$), neither in terms of EORTC QOL-C30 for all three items at both time-points.

IC timing was significantly reduced at T1 ($149.7 \pm 59.8$ s vs $245.9 \pm 93.0$ s; $p = 0.035$) and T2 ($131.2 \pm 42.4$ sec vs $245.9 \pm 93.0$ s; $p = 0.003$), testifying to a good learning curve. Further details are described in Table 2.

**Table 2.** Differences in outcome measures according to the study time-points.

|  | T0 | T1 | T2 | T0-T1 *p*-Value | T1-T2 *p*-Value | T0-T2 *p*-Value |
|---|---|---|---|---|---|---|
| BCRL volume (cm$^3$) |  |  |  |  |  |  |
| Hand | 348.1 ± 99.0 | 293.6 ± 95.2 | 298.6 ± 80.8 | 0.003 * | >0.999 | 0.007 * |
| Forearm | 1654.0 ± 503.8 | 1532.8 ± 433.7 | 1584.7 ± 461.8 | 0.035 * | 0.999 | 0.178 |
| Arm | 1879.3 ± 348.9 | 1695.5 ± 278.9 | 1767.6 ± 337.4 | 0.014 * | 0.472 | 0.472 |
| Total | 3881.4 ± 886.9 | 3521.9 ± 759.2 | 3650.9 ± 835.5 | 0.001 * | 0.472 | 0.102 |
| LLIS | 38.9 ± 10.1 | 34.7 ± 8.1 | 35.1 ± 12.6 | >0.999 | >0.999 | >0.999 |
| EORTC QOL-C30 |  |  |  |  |  |  |
| Function | 86.9 ± 7.2 | 87.4 ± 7.5 | 83.5 ± 9.0 | >0.999 | 0.178 | >0.999 |
| Symptom | 13.7 ± 6.7 | 12.3 ± 4.0 | 15.4 ± 9.3 | >0.999 | 0.716 | >0.999 |
| Global health | 40.7 ± 6.5 | 73.2 ± 16.6 | 31.5 ± 10.0 | 0.716 | >0.999 | 0.297 |
| IC application time (sec) | 245.9 ± 93.0 | 149.7 ± 59.8 | 131.2± 42.4 | 0.035 * | >0.999 | 0.003 * |

Continuous variables are expressed as means ± standard deviations. One-way ANOVA analysis was performed as statistical test. * *p* = <0.05 for significance. Abbreviations: T0 = baseline; T1 = at the end of 2-week treatment; T2 = at the 1-month follow-up visit. BCRL = breast cancer related lymphedema; LLIS = Lymphedema Life Impact Scale; EORTC QOL-C30 = European Organization for Research and Treatment of Cancer Quality of Life questionnaire; IC, inelastic compression.

## 4. Discussion

This proof-of-principle study investigated the safety and tolerability of a new self-adaptive IC device in reducing upper limb volume in women affected by BCRL. To the best of our knowledge, this is the first study investigating the use of IC in BCRL patients suggesting its potential implementation among the CDT interventions. CDT, consisting of MLD, therapeutic exercise, skincare to prevent infection, compression, and bandaging treatment [17,18,31] are considered cornerstones in the rehabilitative management of BCRL women. Moreover, it is well known that therapeutic exercise could be performed with adequate intensity by BC women with a significant reduction of BCRL volume and improvements in terms of upper limb range of motion and muscle strength [32].

In this context, our findings showed that IC could be considered as a safe and well-tolerated intervention by patients and therapists, as confirmed by the high treatment-related satisfaction at the end of the 2-week treatment and the 1-month follow-up. Furthermore, these results were confirmed by the absence of dropouts and side effects, excluding only a low self-limiting discomfort at the hand reported by 1 patient.

To date, data about the safety and tolerability of IC in BCRL patients are lacking. However, it should be noted that the only study in literature was related to the use of velcro-band compression device (Circaid Juxta Lite™) on lower limbs in 30 lymphedema patients without the arterial occlusive disease [33]. The Authors showed that a correct self-application of Velcro bands is effective in maintaining and adequate pressure on the lower limb.

In this scenario, our results provide intriguing data about the safety and tolerability of IC as a new therapeutic strategy in the management of BCRL. Our results show that the self-application of an IC device has a fast-learning curve as confirmed by the significant reduction of the total application time from T0 to T2 (*p* = 0.0029). Furthermore, it should be noted that IC might improve a home-based rehabilitation approach in terms of self-treatment in BCRL patients, reducing hospital length of stay and improving self-treatment and active participation in the rehabilitative treatment.

Regarding upper limb volume modifications, the statistical analysis showed a significant reduction in total upper limb volume after the rehabilitation treatment using IC (*p* = 0.0012). In this scenario, the IC might have a role in enhancing the effects of CDT in reducing BCRL volume in the intensive phase, with intriguing results in terms of HRQoL improvement. We assessed BCRL volume through the CM, considered as the most widely used method in the common clinical practice is still the CM. However, in the recent past, other promising techniques have been proposed for quick volume measurements, including the three-dimensional laser scanner (3DLS).

Considering the small sample size, this pilot study was underpowered to draft any strong conclusion about the effectiveness of IC in upper limb BCRL volume reduction and these preliminary data, although promising, must be confirmed by further studies with larger cohorts.

Lastly, our findings might also suggest intriguing implications in the direct and indirect sanitary costs of BCRL. Previous studies reported that CDT is an expensive intervention that might be carried out only in specialized centers, with an estimated cost of therapy for a single patient that could exceed 2500 EUR per year [34]. Promoting self-treatment, reducing the overall hospital length of stay, device consumption, and health personnel employment could positively impact the overall cost of BCRL therapy. However, at present, a cost analysis is not possible thus, it should be performed in the future on larger samples.

This study was not free from limitations: first, the small sample size that could not lead to any strong conclusions regarding IC efficacy in upper limb volume reduction; second, the absence of a control group, which hinder any true comparison between IC and the MLB; third, the lack of data on muscle strength and functional performance in these patients.

## 5. Conclusions

Taken together, these findings suggest that IC might be considered a safe, well-tolerated, and promising tool in the complex rehabilitative management of BCRL patients, suggesting its potential implementation as a new effective tool in CDT. This proof-of-principle study is the first in literature exploring the safety and tolerability of IC and might be considered as a starting point for future studies evaluating its effectiveness in the context of rehabilitation interventions in BCRL women. Thus, future studies on larger cohorts of patients would be required to ascertain the efficacy and cost-effectiveness of IC in breast cancer precision rehabilitation.

**Author Contributions:** Conceptualization, A.d.S. and M.I.; methodology, A.d.S. and M.I.; formal analysis, A.d.S. and M.I.; investigation, L.L.; resources, C.C.; data curation, A.d.S.; writing—original draft preparation, A.d.S.; writing—review and editing, N.F., E.S., and M.I.; visualization, L.L. and C.C.; supervision, C.C. and M.I. All authors have read and agreed to the published version of the manuscript.

**Funding:** This research received no external funding.

**Institutional Review Board Statement:** The study was conducted according to the guidelines of the Declaration of Helsinki, and approved by the Institutional Review Board.

**Informed Consent Statement:** Informed consent was obtained from all subjects involved in the study.

**Data Availability Statement:** Dataset is available on request.

**Acknowledgments:** The authors would like to thank Laura Colli and Giuliana Guenzi for their support in this work.

**Conflicts of Interest:** The authors declare no conflict of interest. Funders had no role in the design of the study; in the collection, analyses, or interpretation of data; in the writing of the manuscript, or in the decision to publish the results.

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
