# Peer review of "Lymphedema Rehabilitation Using Self-Adaptive Inelastic Compression in Breast Cancer: A Proof-of-Principle Study"

_applsci, doi:10.3390/app11041901_

Round 1
Reviewer 1 Report
This is an important study. My concerns:
Summary – IC timing should be changed to IC application time
Methods – Why the exclusion criterium consists of >2 weeks from the surgery? This should be explained
Further details should be added to the sentence: „…all patients were asked to perform a 2- week home-based rehabilitative treatment, consisting of the same therapeutic exercise learned during the sessions” – MLD was also performed?
What does the abbreviation CM means? How the measurements were performed?
Materials – How the lymphedema was advanced prior the therapy?
Results – Table 1 – the percentage values should not be shown as the numer of participants are small. Percentages of participants should be also removed
Table 2 If data were not normally distributed, why means and standard deviations were shown? This should be corrected
Discussion – The repetition of results data should be avoided
It is not clear whether Circaid is proposed for the intensive phase of CDT or maintenace phase and why? Should it replace elastic compression devices for the whole life? This should be explained and clear message in the conclusion should be given.
Author Response
We are delighted that you have perceived our work positively and provided excellent suggestions that have improved our manuscript substantially. Below you can find a point-by-point response to your comments. In the revised manuscript file, we have highlighted in yellow all changes referring to the Reviewers’ comments and suggestions.
Reviewer #1 comments:
This is an important study.
We thank the reviewer thoroughly for the positive assessment of our study.
- My concerns: Summary – IC timing should be changed to IC application time
According to the Reviewer's recommendation, we have modified the text in the abstract (line 21).
- Methods – Why the exclusion criterium consists of >2 weeks from the surgery? This should be explained
We would like to thank the reviewer for the comment. We now explain the exclusion criterium of >2 weeks from the surgery “page 2, Materials and Methods, Participants section.” (lines 71 and 72)
- Further details should be added to the sentence: „…all patients were asked to perform a 2- week home-based rehabilitative treatment, consisting of the same therapeutic exercise learned during the sessions” – MLD was also performed?
The reviewer is absolutely right. We have better explained this aspect in Materials and Methods, Intervention section by clarifying that MLD has been not performed during the home-based rehabilitative treatment (lines 98-100).
- What does the abbreviation CM mean? How the measurements were performed?
We appreciate the reviewer’s insightful comment. In the Materials and Method section, we now clarify that circumferential measurement (CM) was performed with a tape (1 mm of sensitivity) to measure upper limb circumferences in correspondence of markers made on the skin from wrist to deltoid muscle level, with 5 cm intervals (lines 104-107).
- Materials – How the lymphedema was advanced prior the therapy?
We now clarify this point. Please see line 68.
- Results – Table 1 – the percentage values should not be shown as the numer of participants are small. Percentages of participants should be also removed
As recommended, we have removed percentages in both the Results section and Table 1.
- Table 2 If data were not normally distributed, why means and standard deviations were shown? This should be corrected
We thank the reviewer for outlining this point. We hypothesized that it might be better to report all means and standard deviations. As expressed in the Statistical Analysis subsection, the Shapiro-Wilk test was used to assess the data distribution. However, data distribution non-gaussian, differences in repeated measures were assessed by the Friedman Test (one-way ANOVA) with posthoc Dunnett’s Test as a multiple comparison procedure.
- Discussion – The repetition of results data should be avoided
As suggested, we have removed repetition of results in the Discussion section, leaving only p values when necessary.
- It is not clear whether Circaid is proposed for the intensive phase of CDT or maintenace phase and why? Should it replace elastic compression devices for the whole life? This should be explained and clear message in the conclusion should be given.
We have better clarified that circaid might be proposed in the intensive phase of BCRL rehabilitation (lines 171-177).
Reviewer 2 Report
The authors present a study that may be important in one of the most important complications of cancer patients.However, the article has a limited design and with very superficial data. The sample size is very limited, in each of the analyzed parameters this size does not allow for solid conclusions. The manuscript needs to take into account other parameters related to the quality of life of the patients and especially to carry out a very long-term study. The discussion is limited and should be more concise with the preliminary results shown by the authors. Authors should check their English grammar.
Author Response
We are delighted that you have perceived our work positively and provided excellent suggestions that have improved our manuscript substantially. Below you can find a point-by-point response to your comments. In the revised manuscript file, we have highlighted in yellow all changes referring to the Reviewers’ comments and suggestions.
Reviewer #2 comments:
The authors present a study that may be important in one of the most important complications of cancer patients.
We are grateful for having our manuscript considered as an important study.
- However, the article has a limited design and with very superficial data. The sample size is very limited, in each of the analyzed parameters this size does not allow for solid conclusions. The manuscript needs to take into account other parameters related to the quality of life of the patients and especially to carry out a very long-term study.
We now acknowledge the study limitations in a dedicated paragraph in the Discussion section (lines 176-178).
- The discussion is limited and should be more concise with the preliminary results shown by the authors.
We are grateful to the reviewer for this pertinent observation. To address this point, we improved the Discussion section, as suggested (lines 171-178). Moreover, we clarify that this proof-of-principle study could only suggest that inelastic compression might be considered as a safe, well-tolerated, and promising tool in the complex rehabilitative management of BCRL patients (lines 191-195).
- Authors should check their English grammar.
The manuscript has been revised thoroughly for English language. All changes are highlighted in red.
We hope we have satisfactorily addressed all comments.
We look forward to hearing from you in due course.
Reviewer 3 Report
The article clearly presents the relatively small study on proof-of-principle exploring the safety and tolerability of inelastic compression device. Definitely it is important to test different devices which can help in management of breast cancer related lymphedema.
I have minor recommendation related to more clear description of outcome measures used in the study (lines 111- 115). The measures LLIS and EORTC QLQ-C30 are mentioned and described shortly, but description does not give any idea about the way how results could be interpreted (e.g. min/ max score or categories). The same time description of GPE gives such information (lines 117-118). Perhaps both measures are well known but readers not so familiar with topic about cancer research and lymphedema would not been able to interpret the results for both measures in Table 2 as there are no references for interpretation (just changes over time points).
Reviewer 4 Report
Breast cancer-related lymphedema (BCRL) is a notable post breast cancer issue and the current authors have evaluated the safety and effectiveness of one of the latest treatment tool inelastic compression (IC) to help patients suffering from BCRL. Although the sample size was small and the authors have admitted that, the current study will be served as an important initial study on the safety and efficacy of the tool. Hence I recommend the current study to be published in the current form.
Round 2
Reviewer 1 Report
No further suggestions. Perfect job, congratulations.
Reviewer 2 Report
The authors have not made any of the suggested comments, they have only included the comments as limitations without making new considerations in their study.
The article contains important deficiencies and should not be considered for publication